# The Proper Administration Sequence of Radiotherapy and Anti-Vascular Agent—DMXAA Is Essential to Inhibit the Growth of Melanoma Tumors

**DOI:** 10.3390/cancers13163924

**Published:** 2021-08-04

**Authors:** Alina Drzyzga, Tomasz Cichoń, Justyna Czapla, Magdalena Jarosz-Biej, Ewelina Pilny, Sybilla Matuszczak, Piotr Wojcieszek, Zbigniew Urbaś, Ryszard Smolarczyk

**Affiliations:** 1Center for Translational Research and Molecular Biology of Cancer, Maria Skłodowska-Curie National Research Institute of Oncology, Gliwice Branch, Wybrzeże Armii Krajowej Street 15, 44-102 Gliwice, Poland; Alina.Drzyzga@io.gliwice.pl (A.D.); Tomasz.Cichon@io.gliwice.pl (T.C.); Justyna.Czapla@io.gliwice.pl (J.C.); Magdalena.Jarosz-Biej@io.gliwice.pl (M.J.-B.); Ewelina.Pilny@io.gliwice.pl (E.P.); Sybilla.Matuszczak@io.gliwice.pl (S.M.); 2Brachytherapy Department, Maria Skłodowska-Curie National Research Institute of Oncology, Gliwice Branch, Wybrzeże Armii Krajowej Street 15, 44-102 Gliwice, Poland; Piotr.Wojcieszek@io.gliwice.pl (P.W.); Zbigniew.Urbas@io.gliwice.pl (Z.U.)

**Keywords:** vascular disrupting agents, radiotherapy, brachytherapy, combined anti-cancer therapy, immunotherapy

## Abstract

**Simple Summary:**

The growth of tumors depends on the development of their abnormal vasculature. Targeting tumor blood vessels could be an effective approach to anti-cancer therapy. One of the strategies is targeting existing tumor blood vessels—anti-vascular therapy. Anti-vascular drugs destroy the core of the tumor but leave the rim of viable cells at the periphery of the tumor. These viable cells, so-called tumor rim cells, are resistant to elimination and are responsible for tumor regrowth and therapy failure. Our work aimed to find the correct sequencing of the vascular disrupting agent—DMXAA with radiotherapy (brachytherapy) to improve therapeutic efficacy. Throughout the manuscript, we attempt to explain the importance of immune cells’ activation in such therapy and prove the significance of sequential therapeutic agent administration.

**Abstract:**

Vascular disrupting agents (VDAs), such as DMXAA, effectively destroy tumor blood vessels and cause the formation of large areas of necrosis in the central parts of the tumors. However, the use of VDAs is associated with hypoxia activation and residues of rim cells on the edge of the tumor that are responsible for tumor regrowth. The aim of the study was to combine DMXAA with radiotherapy (brachytherapy) and find the appropriate administration sequence to obtain the maximum synergistic therapeutic effect. We show that the combination in which tumors were irradiated prior to VDAs administration is more effective in murine melanoma growth inhibition than in either of the agents individually or in reverse combination. For the first time, the significance of immune cells’ activation in such a combination is demonstrated. The inhibition of tumor growth is linked to the reduction of tumor blood vessels, the increased infiltration of CD8^+^ cytotoxic T lymphocytes and NK cells and the polarization of macrophages to the cytotoxic M1 phenotype. The reverse combination of therapeutic agents showed no therapeutic effect and even abolished the effect of DMXAA. The combination of brachytherapy and vascular disrupting agent effectively inhibits the growth of melanoma tumors but requires careful planning of the sequence of administration of the agents.

## 1. Introduction

The growth of tumors depends on the development of their vasculature. The progression of small, avascular tumors (1–2 mm^3^) is dependent on the formation their own system of blood vessels. However, tumor blood vessels are immature and chaotic in structure because of their rapid and uncoordinated growth [1]. Additionally, tumor blood vessels leak due to the lack of close adherence between tumor endothelial cells, resulting in loose inter-endothelial cell junctions [2,3]. The consequence of vascular dysfunction is the appearance of hypoxic regions in the tumor microenvironment [4,5,6]. Tumor blood vessels also play an essential role in cancer cells spreading to distant organs [7,8].

Therefore, targeting tumor blood vessels seems to be an effective solution in anti-cancer therapy. Currently, two therapeutic strategies are known: anti-angiogenic therapy, which inhibits the formation of new blood vessels, and anti-vascular therapy, which destroys existing tumor blood vessels.

Both of these strategies have their limitations. Anti-angiogenic drugs activate the alternative mechanism of tumor blood vessels’ formation by releasing pro-angiogenic factors or activating vascular progenitor cells recruited from bone marrow [9,10,11]. For the first time Denekamp described the possibility of tumor blood vessels’ destruction by VDAs [12]. Anti-vascular drugs specifically recognize and destroy blood vessels in tumors [13,14,15,16]. The destruction of tumor blood vessels occurs already 30 min after the administration of VDAs [17]. Tumor endothelial cells are more sensitive to VDAs than endothelial cells present in normal tissues [18]. One of the best-studied anti-vascular agents is DMXAA. Its main mechanism of action is the induction of tumor endothelial cells apoptosis [18,19,20]. DMXAA does not destroy normal tissue because endothelial cells present in normal tissues are more resistant to DMXAA than tumor endothelial cells [18]. The indirect mechanism of DMXAA action is the stimulation of the immune system. The activation of the immune system is based on the activation of mitochondria and an endoplasmic reticulum-associated protein known as a stimulator of interferon genes (STING) [21,22,23].

The use of vascular disrupting agents (VDAs) has been associated with the appearance of extensive areas of hypoxia and necrosis around the damaged blood vessels. These hypoxic areas are observed in the center of the tumor and lead to a reduction in the tumor volume [24,25,26,27]. However, the layer of living cells in the periphery of the tumor, the so-called tumor rim cells, are also observed [27,28]. Tumor rim cells are extremely resistant to eradication [29]. Probably due to surviving peripheral tumor rim cells, tumor shrinkage after VDAs therapy was not observed, especially in human clinical trials [17,24,30].

To improve the therapeutic potential of vascular disrupting agents, it seems to be necessary to associate them with other anti-cancer therapeutic strategies. There are several approaches where VDAs are combined with other anti-cancer therapies such as chemotherapy, radiotherapy, or immunotherapy to improve their effectiveness [13]. Brachytherapy, as one of the radiotherapy techniques, is practical to use when irradiating mice. It is precise and conformal, which means a steep dose gradient between tumor and healthy tissues [31]. It allows designing a repeatable model of radiotherapy treatment in mice.

Our work attempts to find a proper sequence of administration of anti-vascular drug—DMXAA with radiotherapy (brachytherapy) to inhibit murine melanoma growth. Radiotherapy destroys cancer cells, but also affects the tumor microenvironment, mainly by stimulation of immune cells. Cumulative doses of more than 5 Gy also reduce tumor blood vessel density [32]. Fuks et al. showed that a single dose of more than 8–10 Gy and fractionated radiation of 1.8–3 Gy per fraction have strong antivascular effect [33]. In the present study, we have investigated which sequence of administration of these two therapeutic agents is more effective in tumor growth inhibition and explain the reason for the effectiveness of the combination therapy.

## 2. Materials and Methods

### 2.1. Mice and Cell Lines

Mice (8 to 10-week-old, C57Bl/6NCrl) were obtained from Charles River Breeding Laboratories (Wilmington, MA, USA). Experiments on animals were carried out in accordance with standard procedures, with the consent of the Local Ethics Commission of Animal Experiments in Katowice (permission No: 74/2018). Mice were housed in the Maria Sklodowska-Curie National Research Institute of Oncology, Gliwice Branch (Poland) in a pathogen-free facility in SPF standard in a HEPA-filtered Allentown’s IVC System (Allentown Caging Equipment Co, NJ, USA). The animals received a total pathogen-free standard diet (Altromin 1314, Altromin Spezialfutter GmbH and Co. KG, Lage, Germany) and water ad libitum throughout the whole study. Animals were treated in accordance with the recommendations in the Guide for the Care and Use of Laboratory Animals of the National Institutes of Health. Experiments on animals were conducted in accordance with the 3R rule. Murine melanoma B16-F10 cell line (ATCC, Manassas, WV, USA) was cultured in RPMI 1640 (Thermo Fisher Scientific, Waltham, MA, USA) supplemented with 10% heat-inactivated fetal bovine serum (Thermo Fisher Scientific, Waltham, MA, USA). Cell cultures were incubated in standard conditions (37 °C, 5% CO_2_, 95% humidity). Cells were passaged every 3–4 days.

### 2.2. Therapeutic Agent

DMXAA (5,6-dimethylxanthenone-4-acetic acid; Selleckchem, Houston, TX, USA) was injected intraperitoneally at a dose of 25 mg/kg body weight in PBS^-^. Dose and route of administration were selected in accordance with the previous works [34,35,36].

### 2.3. Brachytherapy

Mice with well-developed tumors (60–80 mm^3^) were anesthetized and treated with contact radiotherapy (brachytherapy), as described before [32]. Surface brachytherapy was performed with a dedicated, customized applicator (2 × 2 cm; Freiburg Flap, Elekta, Stockholm, Sweden) placed directly in the tumor area. Surface brachytherapy with such applicator provides minimal dose delivery to surrounding healthy tissue in skin cancers, as described by Skowronek [37]. The dose per fraction of 2 Gy was planned to be specified 2–3 mm from the applicator surface. The dose was adjusted to the tumor thickness to avoid unwanted dose coverage in the organs at risk beyond the tumor. Irradiation was performed in the shielded therapeutic room with a high-dose-rate after-loader equipped with an iridium-192 radioactive source (Microselectron, Nucletron, Veenendaal, Netherlads) in the Brachytherapy Department, Maria Sklodowska-Curie National Research Institute of Oncology, Gliwice Branch (Poland). The fraction delivery time was recalculated with dedicated software every time depending on the source activity (3–10 Ci) and ranged from 30 s to 120 s. The surface applicator was placed and fixed in the tumor area with a cohesive, elastic conforming bandage (Peha-haft, Hartmann, Heidenheim, Germany) before every fraction. The transfer tubes were connected, and irradiation was performed. The applicator was detached immediately after irradiation.

### 2.4. Therapy

Mice were injected subcutaneously (lower flank) with 2 × 10^5^ B16-F10 cells in 100 μL PBS¯. Tumors were measured with calipers and tumor volumes were determined using the formula: volume = width^2^ × length × 0.52. Mice with tumors exceeding 2000 mm^3^ were sacrificed by cervical dislocation. Mice were divided into five treatment groups: Control, DMXAA, Brachytherapy, DMXAA + Brachytherapy, Brachytherapy + DMXAA as shown in the chart below (Scheme 1). In the DMXAA group, ten days after inoculating mice with B16-F10 melanoma cells, mice were treated with a single dose of DMXAA. In the Brachytherapy group, mice were irradiated on the eleventh day after melanoma cells inoculation. Mice irradiation was repeated after four and after next three days. In the DMXAA + Brachytherapy group ten days after melanoma cells inoculation, mice were injected with DMXAA. On the next day and after the next four and three days, mice were irradiated. In the Brachytherapy + DMXAA group ten days after melanoma cells inoculation, mice were irradiated. The following day mice were injected with a DMXAA and after the next four and three days, mice were irradiated.

### 2.5. Histochemical Staining

On the 20th day after melanoma cells inoculation mice from all groups were sacrificed by cervical dislocation and tumors were collected for histochemical analysis. The collected tumors were embedded in OCT (Leica Biosystems, Wetzlar, Germany), frozen in liquid nitrogen and stored at −80 °C until needed. Subsequently, frozen tumors were sectioned into 5 μm slices. The frozen sections were examined histochemically (hematoxylin/eosin staining, Sigma-Aldrich, St. Louis, MO, USA), and the analysis of the specimens was conducted using the Nikon Eclipse 80i microscope (Nikon Instruments Inc., Tokyo, Japan).

### 2.6. Immunohistochemistry

On the 20th day after melanoma cells inoculation mice from all groups were sacrificed by cervical dislocation and tumors were collected for further analysis. The blood vessels in collected tumors were determined by staining frozen sections using antibody directed against CD31 antigen (Abcam; ab7388, 1:50, Cambridge, UK) and subsequently with Alexa Fluor 594 conjugated secondary antibody (Abcam; ab150168, Cambridge, UK). The area occupied by blood vessels was counted with ImageJ software (NIH, Bethesda, MD, USA). To identify the presence and M2 phenotype of macrophages in collected tumors, frozen sections were stained using antibody directed against F4/80 antigen (Abcam; ab6640, 1:100, Cambridge, UK) and subsequently with Alexa Fluor 594 conjugated secondary antibody (Abcam; ab150168, Cambridge, UK) and additionally using antibody directed against CD206 antigen (Abcam; ab64693, 1:100, Cambridge, UK) and subsequently with FITC conjugated secondary antibody (Vector Laboratories; FI-1200, 1:100, Burlingame, CA, USA). The area occupied by macrophages was counted with ImageJ software (NIH, Bethesda, MD, USA). To identify the presence of M1 macrophages in collected tumors, frozen sections were stained using antibody directed against F4/80 antigen (Abcam; ab6640, 1:100, Cambridge, UK) and subsequently with Alexa Fluor 488 conjugated secondary antibody (Abcam; ab150165, Cambridge, UK) and additionally using antibody directed against iNOS antigen (Abcam; ab3523, 1:50, Cambridge, UK) and subsequently with TexasRed conjugated secondary antibody (Vector Laboratories; FI-1200, 1:100, Burlingame, CA, USA). The presence of CD8α T lymphocytes was determined by staining frozen sections using antibody directed against CD8α antigen (Abcam; ab22378, 1:50, Cambridge, UK) and subsequently with Alexa Fluor 594 conjugated secondary antibody (Abcam; ab150168, Cambridge, UK). The presence of NK cells was determined by staining frozen sections using antibody directed against NKp46 antigen (BioLegend #137601, 1:50, San Diego, CA, USA) and subsequently with Alexa Fluor 594 conjugated secondary antibody (Abcam; ab150168, Cambridge, UK). Tumor sections were counterstained with DAPI (Sigma-Aldrich; D8417, St. Louis, MO, USA), and sections were mounted in VECTASHIELD Mounting Medium (Vector Laboratories; H-1000, Burlingame, CA, USA). The numbers of stained cells of each group were counted in 5 randomly chosen fields (magn. 20×) per section in at least 4–5 tumors of each group. Imaging of the fluorescence of the stained sections was performed with the confocal microscope LSM710 (Carl Zeiss Microscopy, Jena, Germany).

### 2.7. Flow Cytometric Analysis 

On the 20th day after melanoma cells inoculation mice from all groups were sacrificed by cervical dislocation and tumors were collected for flow cytometric analysis. To obtain single-cell suspension, excised tumors were minced using scissors, meshed through the 70 µm cell strainer, and washed with the use of PBS¯ supplemented with 1% FBS. Red blood cells were lysed using 0.15 M ammonium chloride (Sigma Aldrich, St. Louis, MO, USA). Lymphocytes isolation from tumor cell suspension was conducted using Lympholyte-M Cell Separation Media (Cedarlane, Burlington, ON, Canada). The subpopulations of T lymphocytes were identified using the following antibodies: FITC-CD45, PE-Cy7-CD4, and APC-CD8 (BioLegend, San Diego, CA, USA). The level of NK cells was determined using the following antibodies: FITC-CD45, PE-CD49b (BioLegend, San Diego, CA, USA). In flow cytometric analyses (BD FACSCanto, BD, Franklin Lakes, NJ, USA), gate dividing negative from positive cells was based on isotype antibody control probes. 7-aminoactinomycin D (7AAD; Thermo Fisher Scientific, Waltham, MA, USA) was used to stain nonviable cells 10 min before running the flow analysis.

### 2.8. Statistics

Results were expressed as mean ± SEM. Statistical analyses were performed using Statistica 12 software (StatSoft). The Shapiro–Wilk test was used to verify the normality of the distribution. For variables with normal distribution, the ANOVA followed by the Tukey’s post hoc test was performed; otherwise, non-parametric testing was carried out (the Kruskal–Wallis followed by the post hoc multiple comparisons of rank sums test). *p* values < 0.05 were considered significant.

## 3. Results

### 3.1. Proper Sequence of Administration of DMXAA and Brachytherapy Inhibits Tumor Growth

The combination of vascular disrupting agent—DMXAA with brachytherapy inhibits the growth of murine melanoma B16-F10 more effectively than either factor alone (Figure 1). However, only the combination where the tumors were irradiated prior to DMXAA administration, was the most effective in tumor growth inhibition. The reverse combination—DMXAA administration prior to brachytherapy—inhibited the growth of tumors but not so effectively. The monotherapy groups—DMXAA and Brachytherapy—inhibited tumors’ growth when the agents were applied up to the 18th day, but afterward, the regrowth of tumors was observed.

### 3.2. The Effect of Combination Therapy on Tumors

After DMXAA administration, large areas of necrosis were observed. After 20 days, in the DMXAA group, tumor regrowth was observed on the edges of the tumors (Figure 2). In the combination group, in which brachytherapy was applied before DMXAA administration, the regions of necrosis were the most extensive. Additionally, after DMXAA administration in monotherapy and combination therapies, infiltration of the immune cells (cells with little cytoplasm and strongly stained with hematoxylin), was observed mainly in the hypoxic regions. There were no differences in necrotic areas and immune cells’ infiltration between the Control and Brachytherapy groups.

### 3.3. The Effect of Combination Therapy on Tumor Blood Vessels Density

Following the respective therapies, tumor blood vessels’ area was determined in murine melanoma tumors. The area of tumor blood vessels was the smallest in the combination group where the brachytherapy was used before the DMXAA administration (Figure 3a). The area of the tumor blood vessels was comparable between the Control, DMXAA, and even the DMXAA + Brachytherapy groups. The higher number of tumor blood vessels in the DMXAA group is due to the presence of a large number of them in the marginal, growing part of the tumors. A non-statistically significant decrease was observed in the Brachytherapy group in comparison to control group. In the Brachytherapy + DMXAA group, the area of the blood vessels was about 50% smaller than in the other groups except for the Brachytherapy group, where the difference was about 25% (Figure 3b).

### 3.4. The Effect of Combination Therapy on Macrophages Polarization

The area of the macrophages in comparison to M1 and M2 phenotype was determined in murine melanoma tumors. The tumor area covered by macrophages (F4/80 positive) was the largest in the Brychytherapy + DMXAA group (Figure 4a,b). The area was seven times larger than in the Control group. The number of macrophages was about five times higher in the DMXAA group than in the Control group. The differences between the Control, Brachytherapy and DMXAA + Brachytherapy groups were not statistically significant. The staining for iNOS positive cells proves that the number of the cytotoxic macrophages was the highest in the combination group where the brachytherapy was used prior to DMXAA in comparison to the other groups (Figure 4c,d).

### 3.5. The Effect of Combination Therapy on CD8^+^ T Lymphocytes Infiltration

The number of infiltrating CD8^+^ cytotoxic T lymphocytes was determined in murine melanoma tumors. The number of CD8 positive cells was the highest in both DMXAA and Brachytherapy + DMXAA groups. The number of CD8 positive cells was about 26 times higher in these groups than in the Control group (Figure 5a). Additionally, the number of CD8 positive cells was 17 times higher in the reverse combination—DMXAA + Brachytherapy than in Control and Brachytherapy groups. The difference between the Brachytherapy and Control groups was not statistically significant (Figure 5b). 

### 3.6. The Effect of Combination Therapy on NK Cells Infiltration

The number of NK cells (NKp46 positive) was determined in murine melanoma tumors (Figure 6a). The number of NK cells was the highest in the Brachytherapy + DMXAA group. The number of NK cells was 17 times higher than in the Control group. The number of NK cells was 10 times higher in the combination group DMXAA + Brachytherapy than in the Control group. The number of NK cells did not significantly differ between the Brachytherapy, DMXAA, and Control groups (Figure 6b).

### 3.7. Immune Cells Infiltration of Tumors after Combined Therapy

The results of immunohistochemistry staining were confirmed by flow cytometry analysis of immune cells composition within tumors (Figure 7a). Analyses from whole tumors confirmed that both the administration of DMXAA and the combination of brachytherapy with DMXAA increased the number of CD8^+^ and CD4^+^ (Figure 7b). An increased number of NK cells was also observed in the Brachytherapy + DMXAA group (Figure 7c). However, the reverse combination, where the DMXAA was administered prior to brachytherapy, did not increase the number of these cells.

## 4. Discussion

Anti-vascular therapy seems to be effective in the treatment of developed solid tumors. The application of the vascular disrupting agents (VDAs) destroys the existing tumor blood vessels, causing large areas of necrosis and hypoxia in the central part of the tumors, which leads to a reduction in tumor volume [20,24,38]. Targeting tumor blood vessels could be more effective than targeting cancer cells because destroying one endothelial cell may lead to the death of many cancer cells [13]. Moreover, targeting tumor blood vessels allows the destruction of hard-to-reach, therapy-resistant cancer cells. The efficacy of VDAs should be tumor type independent. However, some data indicate that well-developed tumors respond better to VDAs than smaller ones [39,40,41]. Reaching the target present on tumor blood vessels is easier, e.g., by the intravenous administration of VDAs than by targeting tumor cells. Drug resistance is not so often observed in endothelial cells as in cancer cells because of greater tumor cell heterogeneity. The use of VDA also reduces the possibility of metastasis by destroying the routes through which cancer cells can migrate.

Currently, many compounds are known to destroy tumor blood vessels. The most well-known are microtubule destabilizing drugs, such as combretastatins, flavonoids with anti-vascular functions such as DMXAA, and tumor vascular-targeted agents such as VEGF- or integrin- targeted fusion peptides or proteins [13]. DMXAA has a strong anticancer effect in mouse tumor models, such as B16.F10 melanoma, 4T1 mammary carcinoma, CT26 colorectal carcinoma [21]. Promising results in preclinical studies involving rodents and in phases I and II of clinical trials have not been confirmed in phase III [17]. The reason for this failure is the sensitivity of DMXAA to murine STING receptors but not human [42,43].Vascular disrupting agents are effective in tumor growth inhibition and even in tumor reduction. However, using such treatment has some limitations. Some of the limitations are their cardiotoxicity and visual disturbance, which are usually transient [24,44]. The second one is that after anti-vascular therapy in the core of tumors, extensive hypoxic areas are observed, but at the edge of the tumors, the remaining dividing neoplastic cells are responsible for the tumor regrowth. Additionally, emerging hypoxia is responsible for the activation of HIF-1α protein in tumors, which in turn is involved in triggering the angiogenesis process and tumor regrowth [45].

The aim of the work was to investigate if the proper sequence of administration of anti-vascular agent—DMXAA and brachytherapy will increase the effectiveness of tumor growth inhibition.

Brachytherapy is a cancer treatment method that has been used for over 100 years [31,46]. In the last two decades, it has undergone significant development with the wide use of image guidance [47]. Brachytherapy uses radiation in the form of a radioactive source or sources placed within or very close to the tumor. It allows reaching a high cancer to normal tissue dose ratio, called conformity [48]. Brachytherapy is used to destroy cancer cells. However, the tumor microenvironment changes are primarily observed by the stimulation of the immune cells. Brachytherapy also increases the infiltration of cytotoxic CD8^+^ T lymphocytes and is responsible for reducing pro-tumorigenic M2 tumor-associated macrophages [32]. Brachytherapy also has an effect on tumor blood vessels. Doses of more than 5 Gy reduce tumor blood vessel density [32]. In our experiments the dose of irradiation was chosen according to the literature, which indicates that radiotherapy in a dose of 2 Gy normalizes tumor blood vessels by reprograming the macrophages to the cytotoxic M1 phenotype [32,49,50]. Firstly, this facilitates the penetration of vascular disrupting drugs into the tumor area and, secondly, it may increase the recruitment of CD8^+^ and CD4^+^ T lymphocytes into the tumor [51]. After such treatment, an additional two brachytherapy applications in a 2 Gy dose should stimulate the immune system to destroy the tumor. It was shown that only fractionated radiotherapy can activate the immune system effectively to treat the tumors [52,53,54].

Two administration sequences of anti-vascular drugs and brachytherapy have been investigated. In the first strategy, brachytherapy was applied 24 h after DMXAA injection. In the second strategy, brachytherapy was applied 24 h before the DMXAA application.

Earlier results indicate that the combination of DMXAA and radiotherapy gives an additive response in tumor growth inhibition [55,56]. Wilson et al. and Murata et al. have shown that irradiation before DMXAA injection gives better results than reverse combination therapy. It was shown that only administration of DMXAA shortly after irradiation results in tumor growth delay and that irradiation after DMXAA injection is less effective [55,56]. However, published results demonstrate only the effect on tumor growth delay in mouse mammary carcinomas and mouse sarcomas without explaining the mechanism of such a process.

Our results indicate that the process of tumor growth inhibition depends on two factors. The first one is the inhibition of tumor blood vessels’ regrowth and the second one is the immune activation.

We have shown that the area occupied by tumor blood vessels was the smallest in the combination group where brachytherapy was applied prior to DMXAA. The area of blood vessels was similar in Control, DMXAA group and in the combination group where the DMXAA was administered prior to brachytherapy. No statistically significant decrease in tumor blood vessel was observed in the Brachytherapy group. No statistically significant difference was also observed between the Brachytherapy and Brachytherapy + DMXAA groups. The lack of significance between the Brachytherapy and Brachytherapy + DMXAA groups may be due to the fact that brachytherapy in a dose of 6 Gy alone reduces the number of tumor blood vessels, and the additional combination with VDAs only slightly reduces their number. Previous works show that doses equal to or above 5 Gy reduce the number of blood vessels in the tumor [32,33]. In the present work, the number of vessels in tumors was analyzed on the 20th day after tumor cells inoculation, where the process of vessel reconstruction could already be observed. It indicates that tumor blood vessels were rebuilt, most of all, after DMXAA monotherapy and in the DMXAA + Brachytherapy groups. The results of other works show that one of the limitations of vascular disrupting agents is the rapid growth of tumor blood vessels after VDAs administration [6,45,57]. The effect of VDAs is related to tumor blood vessels’ destruction and the appearance of hypoxia in the central part of the tumors due to the activation of HIF-1α protein. Hypoxia and the activation of HIF-1α protein in tumors lead to the formation of new blood vessels by the activation of proangiogenic cytokines such as vascular endothelial growth factor (VEGF) and stromal-derived factor 1α (SDF-1α) [6,45,57]. One of the strategies for overcoming this vicious circle associated with tumor blood vessels’ formation is the inhibition of HIF-1α. An example of such an inhibitor is digoxin, which reduces the amount of HIF-1α transcription factor and inhibits the growth of tumors in mice [58]. Our previous results indicate that combining vascular disrupting agents such as DMXAA with Digoxin inhibited tumor re-growth significantly better than monotherapy. Such a combination reduced the number of newly formed vessels and increased the number of macrophages with anti-tumor M1 phenotype, cytotoxic CD8^+^ T lymphocytes and NK cells [34].

Our data also indicate that the effectivity of the proper sequence of administration of the anti-vascular agent and brachytherapy was dependent on the activation of the immune cells. The combinatorial therapy where DMXAA was administered after irradiation is dependent on the polarization of macrophages from the M2 to the M1 phenotype, CD8^+^ cytotoxic T lymphocytes, and NK cells infiltration.

Our results show that the area of macrophages in the tumors was higher in the DMXAA and Brachytherapy + DMXAA groups. No significant differences were observed in the Brachytherapy and DMXAA + Brachytherapy groups compared to the Control group. The combination where DMXAA was applied prior to brachytherapy reduced the number of macrophages in the tumors compared to monotherapy in the DMXAA group. Macrophages play an essential role in obtaining effective cancer therapy. Their polarization from the pro-cancer M2 phenotype to the anti-cancer M1 phenotype is particularly important [59,60,61]. M1 macrophages, besides their phagocytic properties, are responsible for changes in tumor blood vessels and the microenvironment in tumors [62]. Jarosz-Biej et al. indicate that a combination of antiangiogenic drugs and immunostimulatory agents repolarized the M2 macrophages into M1 phenotype. Such repolarization affects the structure of tumor blood vessels. It improves tumor vessel maturation, perfusion and reduces hypoxia, therefore supporting the effect of chemotherapy and leading to tumor growth regression [62].

Our findings also demonstrate that CD8^+^ T lymphocytes are an important population of cells that are responsible for therapeutic effect. We observed that both DMXAA alone and a combination of brachytherapy and DMXAA activated the cytotoxic T lymphocytes in treated mice with B16-F10 melanoma tumors. The reverse application—DMXAA + Brachytherapy—abolishes the effect of CD8 positive cells infiltration. DMXAA, besides its anti-vascular properties, has a robust immunostimulating effect. The activation of immune cells after DMXAA application is mainly related to the protein STING (stimulator of interferon genes). The pathway associated with the STING protein is a relatively new direction of research in cancer therapy [63]. It is well known that immune cells’ activation using STING agonists mediates the innate immune response, which is responsible for the therapeutic effect [21,64]. The authors indicate that the anti-cancer effect of STING protein stimulation was mainly related to strong CD8^+^ T cells infiltration [21,65,66].

In addition to the influx of CD8 lymphocytes, infiltration of NK cells was observed. The infiltration of these cells, which are responsible for tumor growth delay, was the highest in the combination group where the brachytherapy was applied prior to DMXAA injection. It was observed that the number of NK cells was higher in combination therapy even compared to DMXAA alone. The significance of NK cells in tumor growth inhibition was shown in recent papers. Marcus et al. showed that the therapeutic effect of STING agonists is highly related to NK cells activation [67]. It was shown that cGAMP produced by tumor cells triggers the activation of the STING pathway in immune cells from the tumor microenvironment leading to interferons production. Interferons activates NK cells, which are responsible for tumor rejection [67]. The significance of NK cells infiltration after the injection of STING agonists such as cGAMP or DMXAA was also shown in other papers [34,68,69,70]. In all of these papers, it was demonstrated that the infiltration of the NK cells has a tremendous therapeutic value and is mostly responsible for tumor eradication.

It seems reasonable that the anti-tumor response of STING activation depends on immune cells and/or non-immune cells, which should be activated in a strictly planned manner and at the right time of therapy. Zhao et al. demonstrated that only proper sequence of administration of other VDA—combretastatin A4 phosphate with radiotherapy gives a better therapeutic effect. The best effect was observed in rats treated with 5 Gy single dose radiotherapy while oxygen was delivered 24 h prior to combretastatin A4 phosphate (30 mg/kg) administration [71]. It is well known that most immune system cells, such as NK cells, B cells, and T lymphocytes, are radiosensitive [50]. Our results indicate that these cells are mainly responsible for the therapeutic effects. Therefore, after their activation, they could be destroyed by radiation when the sequence of administration is not correct. Another explanation for the observed effect is that the destruction of tumor blood vessels by DMXAA decreases tumor oxygenation and weakens the effect of radiotherapy. It was also shown that hypo-fractionation radiotherapy normalizes the tumor blood vessels [72]. Consequently, after radiotherapy, the administration of DMXAA could give a better therapeutic effect in partly normalized tumor blood vessels. 

In our opinion, the reasonable combination of STING agonists (like DMXAA) and additional immune system stimulation strategies could be beneficial to achieve total tumor eradication. Our results are of great clinical importance because they indicate that only the right combination of radiotherapy and anti-vascular drugs can increase the effectiveness of anti-cancer therapy. The results also indicate that the combination of radiotherapy with immunostimulating compounds, e.g., immune checkpoint inhibitors, should be conducted in such a way as to not abolish their effects. Thus, radiation therapy should be used before or at a strictly scheduled time after immune activation.

## 5. Conclusions

Our results indicate that only the proper sequence of administration of anti-vascular agents and radiotherapy improves tumor growth inhibition. In such a combination, the therapeutic effect is only observed when radiation therapy is given prior to the administration of the vascular disrupting agent. The efficacy of such therapy appears to be dependent on the inhibition of blood vessel regrowth in the tumor and the effective activation of immune cells such as M1 macrophages, cytotoxic T lymphocytes and NK cells.

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
