# Peer review of "The Proper Administration Sequence of Radiotherapy and Anti-Vascular Agent—DMXAA Is Essential to Inhibit the Growth of Melanoma Tumors"

_cancers, 2021, doi:10.3390/cancers13163924_

Round 1

Reviewer 1 Report

Drzyzga et al. investigate the effect of different combination shemes of DMXAA and brachytherapy in a primary melanoma model. A relevant difference on the tumor growth is found in a specific sequence.

The study shows an interesting effect on the combination strategy useful to increase therapy efficacy.

One concern regards the chosen model; it is used a primary melanoma model, growing subcutaneously. This is not exactly close to the clinical condition where this application is likely intended, i.e. metastatic localizations not located in the skin. In fact, in the skin the main treatment is usually surgery. One would have expected Authors would test their DMXAA/radiotherapy combination in a non-skin metastatic localization.

Authors should therefore convincingly discuss how their observations may have relevance in clinical conditions.

A second concern regards the actual differences observed. Namely:

Figure 3B should report the statistical difference between brachytherapy alone and Brachytherapy + DMXAA (third column vs fifth column). Authors  should discuss this difference, which appears not very high.

Figure 5A shows that there is no difference between DMXAA alone and Brachitherapy + DMXAA, indicating that most of the effect observed in Brachytherapy+DMXAA combination is likely related to the DMXAA action.

Line 104: please carefully justify the DMXAA dose chosen and the root of inoculation

Line 110 and line 131 : please carefully justify the brachyterapy dose chosen

Line 236: misspelling for the word “comparision”

Author Response

Thank you for the critical evaluation of our manuscript. All the reviewers comments were included in the revised version of the manuscript. The major corrections are marked in blue in the text. We appreciate reviewers suggestions and we believe that all of the corrections have improved the quality of the manuscript.

The answers for reviewers' suggestions are listed below.

Reviewer #1

One concern regards the chosen model; it is used a primary melanoma model, growing subcutaneously. This is not exactly close to the clinical condition where this application is likely intended, i.e. metastatic localizations not located in the skin. In fact, in the skin the main treatment is usually surgery. One would have expected Authors would test their DMXAA/radiotherapy combination in a non-skin metastatic localization.

  • We choose B16-F10 melanoma model as a standard model for our previous and current therapies. We plan to test proposed combined therapy in additional mice tumor models 4T1 – metastatic breast tumor model and CT26 – colon carcinoma model in a future article. Both tumor models will be performed on BALB/c mice. According to our previous results which indicate that high dose brachytherapy stimulate infiltration of CD8+ lymphocytes, NK cells and polarization of macrophages into M1 phenotype in murine melanoma model, we decided to combine vascular distrupting agents with brachytherapy.

Authors should therefore convincingly discuss how their observations may have relevance in clinical conditions.

  • We discuss at the end of Discussion section (line 487-492) how our observations may have relevance in clinical section. We have mansioned that the results are of great clinical importance because they indicate that only the proper combination of radiotherapy and anti-vascular drugs can increase the effectiveness of anti-cancer therapy. The results also indicate that the combination of radiotherapy with immunostimulating compounds, e.g. immune checkpoint inhibitors should be combined in precisely defined sequence as not to abolish their effects. Thus, radiation therapy should be used before or at a strictly scheduled time after immune activation. Such a strategy should protect from abolition of the immunotherapy effect.

A second concern regards the actual differences observed. Namely:

Figure 3B should report the statistical difference between brachytherapy alone and Brachytherapy + DMXAA (third column vs fifth column). Authors  should discuss this difference, which appears not very high.

  • The lack of statistical significance between the Brachytherapy alone and the Brachytherapy+DMXAA group in Fig. 3B may be due to the fact that brachytherapy alone also reduces tumor blood vessels density, and the additional combination with VDA only slightly reduces their density. Previous works show that doses even or above 5Gy reduce the number of blood vessels in the tumor [Jarosz-Biej et al. 2020; Fuks et al. 2005]. In the literature single-dose radiotherapy in a dose higher than 8-10 Gy or fractionated radiotherapy in low doses – 1.8 – 3 Gy leads to endothelial damage [Fuks et al. 2005]. In presented research, we used brachytherapy in a dose of 6Gy in 3 fractional doses – 2 Gy. In addition, the density of tumor blood vessels was analyzed on 20th day where the process of vessel reconstruction could already be observed.
  • The lack of difference between brachytherapy alone and Brachytherapy + DMXAA have been discussed in the revised version of the publication (line 400-409).  

Figure 5A shows that there is no difference between DMXAA alone and Brachitherapy + DMXAA, indicating that most of the effect observed in Brachytherapy+DMXAA combination is likely related to the DMXAA action.

  • We agree, that the main reason of CD8+ T lymphocytes infiltration in the DMXAA and Brachytherapy + DMXAA groups is related to DMXAA administration. However, for the first time, we have shown that the reverse combination (DMXAA + Brachytherapy) abolishes the effect of CD8+ T lymphocyte stimulation.

Line 104: please carefully justify the DMXAA dose chosen and the root of inoculation

  • DMXAA dose and route of inoculation were chosen based on previous works [Smolarczyk et al. 2019; Wang et al. 2009; Zhao et al. 2002]. This information has been included in the Materials and Methods section. References have been included in the revised version of the article (line 111-112).

Line 110 and line 131 : please carefully justify the brachyterapy dose chosen

  • Radiation dose was chosen according to the literature which indicate that radiotherapy in a dose of 2Gy normalizes tumor blood vessels by reprograming macrophages to cytotoxic M1 phenotype [Klug F, Low-dose irradiation programs 2013; Jarosz-Biej 2019; Jarosz-Biej 2020]. Firstly, this facilitates the penetration of vascular disrupting drugs into the tumor area and, secondly, it may increase recruitment of CD8 + and CD4 + T lymphocytes into the tumor [Frey et al. 2017]. After such a treatment, additional two brachytherapy applications in a 2Gy dose should stimulate the immune system to destroy the tumor. It was shown that only fractionated radiotherapy can activate the immune system effectively to treat the tumors [Hanna et al. 2015; De Keyzer F et al. 2009; Goedegebuure et al., 2019] (line 374-381).

Line 236: misspelling for the word “comparision”

Misspelling was corrected (line 248).

Thanks to the valuable comments from  reviewers and corrections to the work, we hope that you might find our paper interesting and suitable for publishing in Cancers.

Reviewer 2 Report

The authors report a study of melanoma treatment in immune competent mice using combined DMXAA (as a VDA) with local brachytherapy. They test individual treatments as well as combination and specifically consider pre-dosing with DMXAA or applying after 1st dose of radiation. They achieve effective tumor control when DMXAA is given one day after the first of 3 doses of 2Gy radiation. They undertake extensive histology to examine extent of vasculature (CD31), infiltrating immune cells etc. They note shortcoming of VDA treatment alone and discuss the advantages of appropriate combination, specifically using radiation to stimulate an immune response. Indeed they focus on this immune stimulation rather than direct cell kill. This may be because DMXAA failed in clinical trials specifically because it failed to elicit a direct immune response (STING pathway activation) in man, which was an important contributor in studies of mice. Oddly the failed clinical trials with DMXAA are not mentioned.

Specific concerns:

  1. For brachytherapy the authors refer to ref 22. This is from their own lab and did indeed use the same surface brachytherapy applicator, but appears to give few additional details. Vital information includes aspects such as radiation dose distribution. Is there any evidence for a uniform dose to the tumor or is it likely to fall off considerably with depth? How was 2 Gy dose validated? Describe phantom and planning procedure. May be cite Postepy Dermatol Alergol.2015 Oct; 32(5): 362–367.PMCID: PMC4692821 Brachytherapy in the treatment of skin cancer: an overview, Janusz SkowronekDetails are provided in ref 22 for treatment of earlobe of patient, but need to relate to the juxtaposition in applying to a mouse. Truly, how detailed and precise were the treatment plans. What was the approximate duration of radiation delivery?

  1. Discuss the failed clinical trials with DMXAA, e.g., the review of Cite- [1]

  1. There have been extensive previous studies on mechanisms of action. Mechanism with respect to combined therapy may be new in current report.

  1. Some time frames are confusing: line 142 states 20th day post inoculation. Line 179 mentions 18th day of therapy? Fig 1 states day of therapy but zero size suggests day zero is day of implant?

  1. The histology in Fig 2 seems to have lots of tears and artifacts? Fig 2 seems to show tumor for brachy +DMXAA is same size as others, yet Fig1 indicates it should be about 1% volume?

  1. Figure 2 would benefit from scale bars.

  1. How were random regions selected for microscopy?

  1. Try to keep figure legend on same page as Figure, e.g., Fig 3!

  1. To better compare CD31, NK cells and CD8 lymphocytes, was there any attempt to examine similar regions of the tumor? Presumably separate IHC slides?

  1. Fig 7: need alter font to be bigger and legible– it also appears blurry.

  1. General mention of cardiovascular effects of VDAs, but DMXAA apparently had specific effects on vision which should also be mentioned. [2]

  1. Line 80- greater not less than 5Gy? It is correctly stated in Discussion –line 355.

  1. Is 5 Gy the threshold; more often 10Gy- see Fuks et al?

  1. The study would be much more significant if similar data were shown in an additional tumor type, e., RENCA or 4T1 each syngeneic to BLAB/C mice.

  1. What is volume doubling time for these tumors?

  1. How was radiation dose chosen? Based on TCD50? Based on ref23?

  1. Were melanoma cells ever tested for genetic stability? Fungal contamination?

  1. Does DMXAA need a carrier solution?

  1. Suggest adding references to Denekamp [3] who first proposed VDAs and Thorpe [4] who nicely reviewed  a few years ago.

  1. Ref 12 indeed provides a comprehensive perspective on VDA activity, as written by the current authors. To add further breadth I suggest 2 additional pertinent reviews [5] and [6]

  1. In discussing the importance of order of treatment suggest adding reference to the work of Zhao et al, albeit with the other VDA CA4P- see [7]

  1. Line 66-ref 19 seems to discuss cardiotoxicity are rather than tumor shrinkage. Also note due to the surviving peripheral tumor rim, tumors often fail to shrink after VDA treatment alone.

  1. The English distinctly needs linguistic editing; I give a few examples but too extensive for me to list all. e.g.
    1. First paragraph of Intro needs rephrasing.
    2. “VDAs administration is more effective in murine melanoma growth inhibition than each of the” should be “VDA administration is more effective in murine melanoma growth inhibition than either of the”
    3. Line 322 “Application of the vascular disrupting agents (VDAs) destroys existing tumor blood vessels. It causes that large areas of necrosis and hypoxia are observed in the central part of the tumors” better as “Application of vascular disrupting agents (VDAs) destroys existing tumor blood vessels causing large areas of necrosis and hypoxia in the central part of the tumors”

Refs to be added:

  1. Daei Farshchi Adli, A.; Jahanban-Esfahlan, R.; Seidi, K.; Samandari-Rad, S.; Zarghami, N. An overview on Vadimezan (DMXAA): The vascular disrupting agent. Chem Biol Drug Des 2018, 91, 996-1006, doi:10.1111/cbdd.13166.
  2. Jameson, M.B.; Sharp, D.M.; Sissingh, J.I.; Hogg, C.R.; Thompson, P.I.; McKeage, M.J.; Jeffery, M.; Waller, S.; Acton, G.; Green, C.; et al. Transient retinal effects of 5,6-dimethylxanthenone-4-acetic acid (DMXAA, ASA404), an antitumor vascular-disrupting agent in phase I clinical trials. Invest Ophthalmol Vis Sci 2009, 50, 2553-2559, doi:10.1167/iovs.08-2068.
  3. Denekamp, J. Vascular Attack as a Therapeutic Strategy for Cancer. Cancer and Metastasis Reviews 1990, 9, 267-282, doi:Doi 10.1007/Bf00046365.
  4. Thorpe, P.E. Vascular targeting agents as cancer therapeutics. Clinical Cancer Research 2004, 10, 415-427, doi:Doi 10.1158/1078-0432.Ccr-0642-03.
  5. Liu, L.; O’Kelly, D.; Schuetze, R.; Carlson, G.; Zhou, H.; Trawick, M.L.; Pinney, K.G.; Mason, R.P. Non-Invasive Evaluation of Acute Effects of Tubulin Binding Agents: A Review of Imaging Vascular Disruption in Tumors. Molecules 2021, 26, 2551.
  6. Nepali, K.; Ojha, R.; Lee, H.Y.; Liou, J.P. Early investigational tubulin inhibitors as novel cancer therapeutics. Expert Opin Investig Drugs 2016, 25, 917-936, doi:10.1080/13543784.2016.1189901.
  7. Zhao, D.; Chang, C.-H.; Kim, J.G.; Liu, H.; Mason, R.P. In vivo near-infrared spectroscopy and MRI monitoring of tumor response to Combretastatin A4 phosphate correlated with therapeutic outcome. Int. J. Radiat. Oncol. Biol. Phys. 2011, 80 574-581 doi:doi:10.1016/j.ijrobp.2010.12.028; PMID: 21345614

Author Response

Thank you for the critical evaluation of our manuscript. All the reviewers comments were included in the revised version of the manuscript. The major corrections are marked in blue in the text. We appreciate reviewers suggestions and we believe that all of the corrections have improved the quality of the manuscript.

The answers for reviewers' suggestions are listed below.

Indeed they focus on this immune stimulation rather than direct cell kill. This may be because DMXAA failed in clinical trials specifically because it failed to elicit a direct immune response (STING pathway activation) in man, which was an important contributor in studies of mice. Oddly the failed clinical trials with DMXAA are not mentioned. 

  • We included information about failed clinical trials with DMXAA. We have discussed DMXAA has a strong anticancer effect in mouse models of tumors like: B16.F10 melanoma, 4T1 mammary carcinoma, CT26 colorectal carcinoma [Corrales et al., 2015]. Promising results in mouse models and phases I and II of clinical trials have not been confirmed in phase III involving humans [Daei Farshchi Adli et al. 2018]. The reason for the clinical trials failure is the sensitivity of DMXAA to murine STING receptors but not to human species [Conlon et al. 2013; Kim et al. 2013] (line 349-353).

Specific concerns: 

  1. For brachytherapy the authors refer to ref 22. This is from their own lab and did indeed use the same surface brachytherapy applicator, but appears to give few additional details. Vital information includes aspects such as radiation dose distribution. Is there any evidence for a uniform dose to the tumor or is it likely to fall off considerably with depth? How was 2 Gy dose validated? Describe phantom and planning procedure. May be cite Postepy Dermatol Alergol.2015 Oct; 32(5): 362–367.PMCID: PMC4692821 Brachytherapy in the treatment of skin cancer: an overview, Janusz SkowronekDetails are provided in ref 22 for treatment of earlobe of patient, but need to relate to the juxtaposition in applying to a mouse. Truly, how detailed and precise were the treatment plans. What was the approximate duration of radiation delivery?
  • Brachytherapy is a method of irradiation that is highly precise and repeatable [Dickhoff et al. 2021; Thomadsen et al. 2016]. In our experiments we used comercial applicators that are used in human practice. The applicators were checked on a standard phantom as part of quality assurance. Standard 2D planning was used. The dose per fraction of 2 Gy was planned to be specified 2–3mm from the applicator surface. The dose was adjusted to the tumor thickness to avoid unwanted dose coverage in the organs at risk beyond the tumor. Surface brachytherapy with such applicator provides minimal dose delivery to surrounding healthy tissue in skin cancers as described by Skowronek [Skowronek et al. 2015]. The fraction delivery time was recalculated with dedicated software every time depending on the source activity (3–10 Ci) and ranged from 30s to 120s (line 127).
  • Citation of Skowronek et al. was included. Information about fraction delivery time was supplemented (line 118-120).
  1. Discuss the failed clinical trials with DMXAA, e.g., the review of Cite- [1]
  • We included information about failed clinical trials with DMXAA. We have discussed DMXAA has a strong anticancer effect in mouse models of tumors like: B16.F10 melanoma, 4T1 mammary carcinoma, CT26 colorectal carcinoma [Corrales et al., 2015]. Promising results in mouse models and phases I and II of clinical trials have not been confirmed in phase III involving humans [Daei Farshchi Adli et al. 2018]. The reason for the clinical trials failure is the sensitivity of DMXAA to murine STING receptors but not to human species [Conlon et al. 2013; Kim et al. 2013]. Proposed reference: Daei Farshchi Adli et al. 2018 was included (line 349-353).
  1. There have been extensive previous studies on mechanisms of action. Mechanism with respect to combined therapy may be new in current report. 
  • In the present work we mainly focused on the influence of proposed combined therapy on the immune cells response. We have found that immune cells activation is responsible for the observed effect mainly by NK cells activation. Studies on mechanism of action would be beneficial for broaden current knowledge. However, we actually did not examine the mechanism of action of such a combined therapy, therefore we removed word “mechanism” from presented paper. For the first time, we have shown that reverse combination (DMXAA + Brachytherapy) abolishes the effect of CD8+ T lymphocyte infiltration.

  1. Some time frames are confusing: line 142 states 20th day post inoculation. Line 179 mentions 18th day of therapy? Fig 1 states day of therapy but zero size suggests day zero is day of implant? 

  • 18th day of therapy in line 179 was a mistake, mice for each experiment were sacrificed 20th day post inoculation. The mistake was corrected in the revised version of the publication. We have also added the information of tumor collection time in the Material and methods section and in the figure legends (line 190-191).
  • We agree that inscription “day of therapy” on Fig. 1 could be misleading because zero on X axis indicates day of implantation. Therefore we changed the graph, x-axis numeration starts from the 10th day - it means 10 days after tumor cells inoculation; the first day of therapy (line 221).
  1. The histology in Fig 2 seems to have lots of tears and artifacts? Fig 2 seems to show tumor for brachy +DMXAA is same size as others, yet Fig1 indicates it should be about 1% volume? 
  • There is lots of artifacts in Fig. 2 because tumors were embedded in OCT and then frozen in liquid nitrogen and cut with the use of cryotom – it is a standard and optimized procedure for our immunostainings (macrophages, endothelial cells etc.). We are aware that for hematoxylin/eosin staining tumors should be fixed in formalin and embedded in paraffin. However, according to the ethic committee agreement and 3R rules, we tried to limit the number of mice used in the experiments.
  • Tumor from the Brachytherapy+DMXAA group seems to be in the same size as others, however, scale bar indicate the true size of tumors. Additionally, images from the Brachytherapy + DMXAA group showed mainly the regions of necrosis and infiltration of immune cells than tumor tissue (line 236).
  1. Figure 2 would benefit from scale bars. 
  • The visibility of the scale bars has been improved (line 236).
  1. How were random regions selected for microscopy? 
  • Whole tumor section was firstly investigated under confocal microscope. Then, image sharpness was set on nuclei using DAPI staining, and after that the image was taken.
  1. Try to keep figure legend on same page as Figure, e.g., Fig 3! 
  • We checked final version of paper and figures were kept with legend on the same page in the revised version of the article.
  1. To better compare CD31, NK cells and CD8 lymphocytes, was there any attempt to examine similar regions of the tumor? Presumably separate IHC slides? 
  • We did not perform different staining of similar regions of the tumor (separate IHC slides). However, we stained slides from the same tumor for each kind of immunostainings. Tumor tissues were cut for 5um thick sections. Serial sections with 5um interval from the same tumor were created and the successive sections were stained against CD31, NK, CD8 antigens.
  1. Fig 7: need alter font to be bigger and legible– it also appears blurry. 
  • We agree that font in Fig.7 were to small and the graphs were non legible. We have changed them to be more legible (line 323-325).
  1. General mention of cardiovascular effects of VDAs, but DMXAA apparently had specific effects on vision which should also be mentioned. [2] 
  • We have updated the description of VDAs accordingly to the Reviewer suggestion (line 355-356).
  1. Line 80- greater not less than 5Gy? It is correctly stated in Discussion –line 355. 
  • There was a mistake in line 80. We have corrected it to be consistent with Discussion and have cited respective paper (line 84).
  1. Is 5 Gy the threshold; more often 10Gy- see Fuks et al? 
  • Our statement about 5 Gy is based on our previous researches [Jarosz-Biej et al. 2020], where we have observed that doses even and above 5 Gy significantly reduce number of tumor blood vessels. However, we cannot state that 5 Gy is the threshold. After reviewing with the paper of Fuks et al. we have updated Introduction and Discussion paragraphs. We indicated that according to Fuks et al., single dose of more than 8-10Gy and fractionated radiation of 1,8-3Gy per fraction have strong antivascular effect (lines 85-86; 406-407).

  1. The study would be much more significant if similar data were shown in an additional tumor type, e., RENCA or 4T1 each syngeneic to BLAB/C mice. 

  • We agree that the examination on additional tumor models would improve the study. We plan to test proposed combined therapy on others than B16-F10 mice models, which are 4T1 – breast tumor model and CT26 – colon carcinoma model in the future paper.
  1. What is volume doubling time for these tumors? 
  • Volume doubling time calculated with the Schwartz (Schwartz M; A biomathematical approach to clinical tumor growth. Cancer. 1961; 14:1272–1294) equation T*tlog(2) / log(V/Vo) (where T* is the tumor doubling time in days, t is the time in days between volume measurements, V is the current tumor volume, and Vo is the initial tumor volume). For control tumors volume doubling time is of around 2,6 days. For DMXAA is of around 3,8 days. For Brachytherapy tumors volume doubling time is of around 2,8 days and for DMXAA + Brachytherapy tumors volume doubling time is of around 4,6 days. While for Brachytherapy+DMXAA tumors after 14 days post inoculation volume doubling time was impossible to calculate, because initial volume was higher than current.
  1. How was radiation dose chosen? Based on TCD50? Based on ref23? 
  • Radiation dose was chosen according to the literature which indicate that radiotherapy in a dose of 2Gy normalizes tumor blood vessels by reprograming macrophages to cytotoxic M1 phenotype [Klug F, Low-dose irradiation programs 2013; Jarosz-Biej 2019; Jarosz-Biej 2020]. Firstly, this facilitates the penetration of vascular disrupting drugs into the tumor area and, secondly, it may increase recruitment of CD8 + and CD4 + T lymphocytes into the tumor [Frey et al. 2017]. After such a treatment, additional two brachytherapy applications in a 2Gy dose should stimulate the immune system to destroy the tumor. It was shown that only fractionated radiotherapy can activate the immune system effectively to treat the tumors [Hanna et al. 2015; De Keyzer F et al. 2009; Goedegebuure et al., 2019] (line 374-381).

  1. Were melanoma cells ever tested for genetic stability? Fungal contamination? 
  • B16-F10 cell line was purchased from ATCC and used up to eighth passage. In such short time there should not be observed genetic instability. We did not check cell line for fungal contamination, however cells were cultured in sterile conditions without any fungal contamination visible under the microscope.
  1. Does DMXAA need a carrier solution?
  • DMXAA was injected intraperitoneally at a dose of 25 mg/kg body weight in PBS-. The dose and route of administration were chosen in accordance with previous works [Smolarczyk et al. 2019; Wang et al. 2009; Zhao et al. 2002]. We have updated description about carrier solution in the revised version of the publication (line 111-112 line).
  1. Suggest adding references to Denekamp [3] who first proposed VDAs and Thorpe [4] who nicely reviewed  a few years ago. 
  • We agree that these references are consistent with our paper and they will breadth and improve it. We have added references into final version of publication (line 55-58).
  1. Ref 12 indeed provides a comprehensive perspective on VDA activity, as written by the current authors. To add further breadth I suggest 2 additional pertinent reviews [5] and [6] 
  • Indeed, these references can raise the level of our article. We have added references in the revised version of publication (line 57-58).
  1. In discussing the importance of order of treatment suggest adding reference to the work of Zhao et al, albeit with the other VDA CA4P- see [7]
  • We discuss the results of Zhao et al. in the Discussion section and we have added the reference according to Reviewer suggestion (line 472-476).
  1. Line 66-ref 19 seems to discuss cardiotoxicity are rather than tumor shrinkage. Also note due to the surviving peripheral tumor rim, tumors often fail to shrink after VDA treatment alone.
  • The paragraph was changed. We have written that probably due to surviving peripheral tumor rim cells, tumor shrinkage after VDAs therapy was not observed, especially in human clinical trials. Additionally, we have updated references (line 72-73).

  1. The English distinctly needs linguistic editing; I give a few examples but too extensive for me to list all. e.g.
    1. First paragraph of Intro needs rephrasing.
    2. “VDAs administration is more effective in murine melanoma growth inhibition than each of the” should be “VDA administration is more effective in murine melanoma growth inhibition than either of the”
    3. Line 322 “Application of the vascular disrupting agents (VDAs) destroys existing tumor blood vessels. It causes that large areas of necrosis and hypoxia are observed in the central part of the tumors” better as “Application of vascular disrupting agents (VDAs) destroys existing tumor blood vessels causing large areas of necrosis and hypoxia in the central part of the tumors” 

  • Linguistic corrections have been made by an English-speaking scientist proficient in English. His personal data was added in “Acknowledgments” paragraph (line 512-513).
  1. All recommended references have been added to the text of the publication.

Thanks to the valuable comments from  reviewers and corrections to the work, we hope that you might find our paper interesting and suitable for publishing in Cancers.

Round 2

Reviewer 1 Report

Authors addressed all issues

Author Response

English style and language has been checked and corrected.